# Choosing between city and suburb: How urbanization shapes graduates' housing preferences

Shuxia Li ⓘ*, Zeqin Ye

Concord University College Fujian Normal University, Fuzhou, Fujian, China

* lishuxia2205@163.com

## Abstract

This study explores the housing selection mechanism of college graduates between urban and suburban areas under the background of accelerated global urbanization and regional differentiation, focusing on how they balance multiple factors such as employment opportunities, transportation convenience, living costs, career development, residential satisfaction, and family support under conditions of incomplete information and resource constraints. Research the integration of bounded rationality theory and consumer satisfaction theory to construct an analytical framework, collect data through questionnaire surveys, and use structural equation modeling for empirical testing. Research has found that external environmental factors (employment opportunities, transportation convenience, living costs, and work environment) significantly affect housing choices through direct and indirect pathways, with living costs and career planning showing a positive driving effect, while residential satisfaction plays a key mediating role. It is particularly noteworthy that family economic support not only directly affects housing decisions, but also significantly regulates the strength of the impact of living costs and employment opportunities. The research results not only expand the application of behavioral decision-making theory in the housing field but also provide important basis for government departments to optimize regional resource allocation, improve transportation networks, and formulate targeted housing policies. At the same time, it also provides practical reference for real estate developers' market positioning and university employment guidance services. The comprehensive analysis framework established in this study also lays the methodological foundation for future largsssse-scale studies that include more dynamic variables.

## 1. Introduction

The accelerating pace of globalization and urbanization has precipitated profound transformations in urban functions and regional differentiation. According to the United Nations World Urbanization Prospects report, 55% of the global population

**Data availability statement:** The relevant data are available in a Mendeley repository with the following DOI: [10.17632/s28bvjhksj.1] (https://doi.org/10.17632/s28bvjhksj.1).

**Funding:** The author(s) received no specific funding for this work.

**Competing interests:** The authors have declared that no competing interests exist.

currently resides in urban areas, with projections indicating this figure will rise to 68% by 2050 (United Nations, 2018). China exemplifies this trend, with its urbanization rate reaching in 2024, reflecting both the expansion of megacities like Beijing and Shanghai and the accompanying concentration of resources and rising living costs [1]. Notably, housing prices in 23 of 70 major Chinese cities increased in 2024, with Nanjing, Chengdu, and Sanya each experiencing a 0.6% rise [1]. This spatial disparity underscores the fundamental trade-off between urban centers offering "high-cost, high-opportunity" environments and suburban areas characterized by lower costs but relatively limited prospects. For recent college graduates, this dichotomy presents a critical dilemma. While urban cores provide access to premium employment opportunities that facilitate career advancement, they simultaneously impose substantial financial burdens through exorbitant rents, living expenses, and commuting pressures. Consequently, graduates must navigate complex trade-offs between opportunity-rich yet costly urban locales and more affordable but opportunity-constrained suburban alternatives. This decision-making process not only reflects broader macroeconomic shifts in global economic restructuring and uneven regional development but also carries significant implications for urban planning, labor market dynamics, and graduates' long-term career trajectories and quality of life.

Theoretical foundations for understanding such decisions originate with Simon's (1957) [2] seminal work on bounded rationality, which posits that individuals, constrained by limited information and cognitive resources, employ heuristic strategies to balance risks and rewards when evaluating alternatives. While this framework has been widely applied in management science (March, 1994) and increasingly in urban studies, extant research has paid insufficient attention to the role of subjective evaluations—particularly residential satisfaction—in housing decisions. Although satisfaction metrics have been extensively utilized in consumer behavior [3] and organizational research, their application to residential choice remains underdeveloped [4] . The accelerating pace of urbanization has reshaped housing markets globally, creating a critical dilemma for recent graduates: choosing between high-cost, high-opportunity urban centers and more affordable yet resource-constrained suburbs. While prior studies have examined macroeconomic drivers of urban migration [5,6], three key gaps persist: limited integration of graduates' subjective evaluations (e.g., residential satisfaction) with objective economic constraints; insufficient attention to familial support as a moderating factor and outdated statistical analyses of cost-opportunity trade-offs in emerging economies [7]. This study addresses this gap by developing an integrated theoretical model that incorporates eight key constructs: employment opportunities, transportation accessibility, living costs, work environment, residential satisfaction, housing choice, family economic support, and personal career planning. Building on prospect theory [8] , we examine how individuals assess costs and benefits under conditions of uncertainty, particularly in the context of urban-suburban residential decisions. Our framework identifies two primary pathways of influence: The first one is direct effects, where personal career planning and employment opportunities exert positive influences on housing choices [9]; and another is interactive effects, wherein transportation accessibility not only enhances

access to employment but also indirectly mitigates living cost pressures through reduced commuting expenses. Crucially, we introduce family economic support as a moderating variable, examining its role in shaping housing decisions across varying levels of economic constraint and employment opportunity [10] .

This study integrates core concepts such as employment opportunities, transportation convenience, living costs, work environment, residential satisfaction, housing choices, family economic support, and personal career planning. Supported by the theories of "opportunity cost" and "resource allocation", a multidimensional and dynamic interactive theoretical framework is constructed to systematically reveal the complex internal connections between various variables in the process of choosing residential areas in urban centers and suburbs. This study advances theoretical understanding by integrating bounded rationality, prospect theory, and satisfaction research into a comprehensive framework for analyzing urban housing decisions. Moreover, it provides novel empirical evidence regarding the mediating role of residential satisfaction and the moderating effect of family support in graduate housing choices. This study offers practical insights for policymakers seeking to optimize urban resource allocation through targeted interventions such as housing subsidies, transportation improvements, and employment support programs. By elucidating the complex interplay of objective constraints and subjective evaluations in residential decision-making, this research contributes to more nuanced approaches to urban development and human capital management in an era of rapid urbanization.

## 2. Theoretical background

### 2.1. Theoretical foundations of satisfaction in housing decision theory

Housing decision-making is a complex cognitive and behavioral process. It involves interactions among cognitive, emotional, economic, and social factors. Bounded rationality theory [11] posits that individuals often adopt a satisficing solution rather than a fully optimal one. They then trade off economic cost, employment opportunities, and quality of life. Joshanloo [12] further outlines information processing, preference formation, and decision evaluation. He emphasizes the role of subjective emotions in final judgments. Prospect theory [13] adds that under uncertainty, trade-offs among housing cost, commuting expenses, and future job potential are not linear. Instead, they display reference dependence and nonlinear risk preferences. In consumer behavior research, Bhuiyan et al. [14] emphasize that satisfaction depends on both objective conditions and on expectation management and actual experience. Korkidakis et al. [15] argue that discrepancies between reference points, expected outcomes, and actual outcomes determine decision satisfaction. Recently, Rodriguez-Garcia et al. [16] introduced family economic support as a moderator in housing decision models. They examined how family background influences housing choice and satisfaction under varying economic pressures and employment opportunities. This integrated framework connects individual decision-making processes with broader urban housing market dynamics. It accounts for both structural constraints and personal adaptation strategies, offering a comprehensive approach to understanding graduate housing choices in competitive urban environments. The theory bridges economic models with behavioral insights, providing a robust foundation for analyzing how graduates navigate the challenges of housing affordability and employment access.

### 2.2. Resource allocation and trade-off mechanisms from an opportunity cost perspective

Resource allocation and trade-off mechanisms are crucial to housing decisions from an opportunity cost perspective. Opportunity cost refers to benefits forgone under limited resources to achieve a goal. It includes direct economic costs, time, emotional effort, and future returns [17]. This tension is most pronounced in choices between urban centers and suburbs. City centers offer high-quality jobs, robust public services, and rich social networks. However, rents, living expenses, and commute times also rise sharply [18]. In contrast, suburbs have lower housing costs but fewer job opportunities, weaker agglomeration effects, and limited transit [19]. Prospect theory suggests that under uncertainty, individuals avoid losses. They include hidden costs—time, emotional strain, and lost future opportunities in decisions. As a result, they may

favor lower-cost options over high-risk, high-return ones [20]. Resource allocation theory emphasizes rational distribution of money, time, and effort under scarcity. The goal is to maximize overall benefit. When choosing housing, recent graduates compare rent, commute costs, and living expenses. They also weigh social networks, career development opportunities, and future income potential [21]. Additionally, family economic support serves as a key moderator. It can ease individual burdens under varied economic pressures and job markets. This influence alters final housing choices [22]. Drawing on these theories, we develop a multidimensional model that integrates direct costs, emotional costs, and future returns. This framework systematically reveals how graduates allocate resources and make trade-offs in urban versus suburban housing choices. It offers a robust theoretical basis for urban planning, public resource optimization, and evidence-based decision making by graduates.

## 3. Hypotheses

### 3.1. Conceptual model

This study proposes a research model (illustrated in Fig 1) that examines the multifaceted factors influencing urban residents' housing choices, with a focus on the interplay between economic stability, personal aspirations, and practical living conditions. The framework identifies eight core dimensions: employment opportunities, family economic support, convenience transportation, satisfaction, living cost, work environment, housing choice, and personal career planning. These dimensions collectively shape individuals' decisions in selecting residential locations, reflecting both objective constraints and subjective preferences. The model positions employment opportunities and family economic support as foundational economic drivers, directly affecting residents' ability to afford housing. Convenience transportation and work environment represent practical considerations, where accessibility and workplace proximity significantly influence daily life quality [5]. Satisfaction and living cost act as balancing factors, mediating

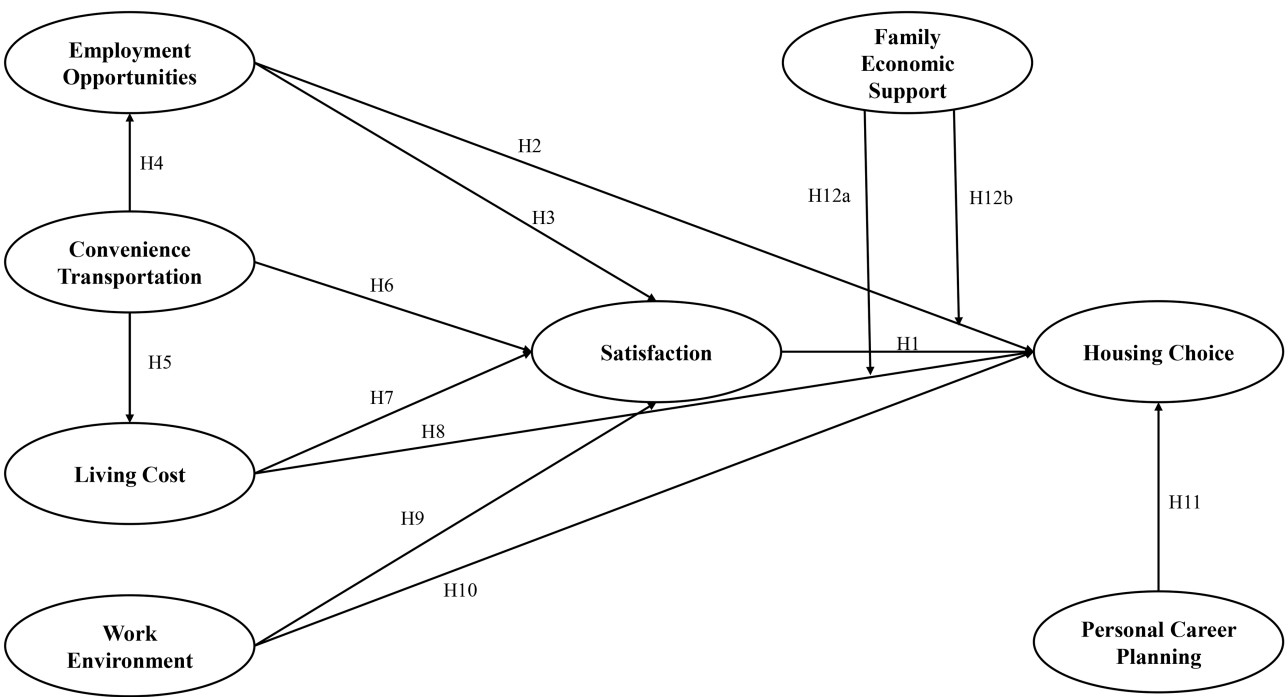

**Fig 1. Research model.**

the trade-offs between affordability and lifestyle expectations [7]. Housing choice emerges as the central outcome, synthesized from these competing priorities. A critical moderating role is assigned to personal career planning, which alters the weight individuals assign to other factors. For early-career professionals, employment opportunities and career growth may dominate, while families might prioritize family economic support and living cost. The model integrates theories of rational choice and quality-of-life appraisal, highlighting how housing decisions are rarely linear but instead involve dynamic negotiations among economic realities, personal goals, and environmental conditions. By delineating these pathways, the framework advances a holistic understanding of urban housing dynamics, offering insights for policymakers and urban planners aiming to align housing supply with diverse resident needs. It underscores the importance of tailored interventions, such as transit-oriented development for commuters or affordable housing programs for low-income families to address the heterogeneous priorities within urban populations. The conceptual model is shown as Fig 1.

### 3.2. Satisfaction

Satisfaction in this study is residential satisfaction, which reflects individuals' subjective evaluation of their living environment's congruence with personal needs and expectations [23]. Grounded in consumer satisfaction theory, RS has been empirically validated as a pivotal mediator between environmental stimuli and housing decisions [24,25]. High RS typically stems from optimal perceived utility across dimensions including spatial functionality, neighborhood quality, and cost-performance ratios, which collectively enhance the propensity to sustain or replicate current housing choices [26]. The bounded rationality perspective further posits that under information asymmetry, satisfaction acts as a heuristic shortcut for decision-making [27,28]. Synthesizing these theoretical lenses, this study formulates the following hypothesis.

H1: Residential satisfaction is positively related to housing choice.

### 3.3. Employment opportunities

Employment opportunities (EO) are a fundamental determinant of housing decisions, particularly for college graduates undergoing urban or suburban transformation [29,30]. Based on the spatial mismatch theory [31], employment opportunities affect housing preferences through a dual mechanism: proximity to high-quality jobs reduces commuting costs, increases income potential [32], and entry into dynamic labor markets accelerates career growth [33]. There is further theory suggesting that when making housing choices under incomplete information, graduates prioritize employment opportunities and use job availability as a significant heuristic method [34]. Empirical research consistently shows that even after controlling for housing costs, metropolitan areas with a 10% higher employment density will attract 15–22% of graduate immigrants [35,36]. Meanwhile, EO has an indirect impact through resident satisfaction (RS). The accessibility of work promotes psychological security and self actualization [37], which translates into higher neighborhood attachment [38]. Hence, the hypotheses formulated as follows.

H2: Employment opportunities are positively related to housing choice.

H3: Employment opportunities are positively related to satisfaction.

### 3.4. Transportation accessibility

Transportation accessibility (TA), defined as the ease of reaching desired destinations through available transport networks [39], serves as a critical determinant in graduates' housing decisions. Regional structure and environmental factors shape basic regional functions and living conditions. They do so through multiple dimensions: transport networks, economic activity density, and community resource distribution. These factors also profoundly affect graduates' employment opportunities and living costs [40]. Transportation accessibility is a core aspect of regional structure. It shortens commute distances and times. This increases graduates' access to a broader job market and concentrates employment opportunities [41]. However, areas with efficient transport often have higher rents and living expenses. This exacerbates financial

burdens and lowers residents' actual quality of life [42]. Therefore, graduates must carefully weigh high opportunities against high costs when choosing housing. Meanwhile, family economic support acts as an important moderator. Hence, the following hypotheses are constructed.

H4: Transportation accessibility is positively related to employment opportunities.

H5: Transportation accessibility is positively related to living costs.

H6: Transportation accessibility is positively related to satisfaction.

### 3.5. Living costs

The cost of living (LC), including housing expenses, daily consumption, and service prices [43], is a key factor in graduates' housing choices. According to the pressure resource theory [44], LC affects housing preferences through two offsetting mechanisms: financial pressure and compensatory choice. High LC depletes disposable income and reduces satisfaction [45]. The cost of living affects financial burden and disposable income. Excessive costs can lead to stress and anxiety. Moderate costs allow for more freedom and better quality of life [46].

H7: Living costs are negatively related to satisfaction. Hence, the following hypotheses are constructed.

H8: Living cost is related to housing choice.

### 3.6. Work environment

The work environment (WE), including office facilities, corporate culture, team atmosphere, and career development opportunities, can enhance life satisfaction. It improves emotional stability and happiness [47]. Based on the job demand resource theory [48], positive workplace experiences increase resident satisfaction, and proximity premiums can make housing near ideal workplaces more attractive. A good working environment can partially offset the impact of high living costs and indirectly affect housing choices [49]. Hence, the hypotheses are formulated as follows.

H9: Work environment is positively related to satisfaction.

H10: Work environment is positively related to housing choice.

### 3.7. Personal career planning

Personal career planning (PCP), defined as the systematic process of setting professional goals and mapping pathways to achieve them [50] , serves as a critical cognitive driver in graduates' residential choices. Grounded in human capital theory [51] , PCP influences housing selection through strategic location investment where residential decisions are consciously aligned with long-term career trajectories. Graduates with well-developed PCP use career optimization as a dominant heuristic when navigating complex housing trade-offs [52–54]. Hence, the hypothesis is formulated as follows.

H11: Personal career planning is positively related to housing choice.

### 3.8. Family economic support

Family economic support (FES), defined as the financial resources provided by parents or relatives to facilitate independent living [55] . It can substantially ease the pressure of high living costs [56]. When family support is sufficient, graduates tend to choose high-cost areas with abundant job resources and strong public services. When support is limited, they may opt for more conservative choices. These choices balance quality job opportunities against manageable living costs. This moderating effect reduces psychological burdens during decision-making. It also optimizes urban housing resource allocation at the macro level. Hence, the following hypotheses are constructed [57,58].

H12a: Family economic support moderates the relationship between living costs and housing choice.

H12b: Family economic support moderates the relationship between employment opportunities and housing choice.

# 4. Methods

## 4.1. Questionnaire item sources

The questionnaire was designed to explore the multidimensional factors that influence college graduates' urban housing decisions. Respondents must hold at least a bachelor's degree and have graduated within the past five years (to capture recent housing decisions while allowing for a sufficiently large sample). Participants were recruited from 21 universities across the five cities, stratified by institution type (e.g., elite "985" universities, vocational colleges) to ensure socioeconomic diversity. Measurement items were drawn from relevant literature (see Table 1) and adapted to field requirements. This approach ensures both theoretical rigor and practical feasibility. Variables were grouped into three categories. Category 1: Regional environment factors (employment opportunities, transportation accessibility, living costs, work environment). Category 2: Residential outcome variables (housing choice, life satisfaction). Category 3: Individual and family background factors (family economic support, personal career planning). All data were anonymized and used solely for statistical analysis and academic dissemination. Respondent privacy was maintained. Out of 400 distributed questionnaires (200 online and 200 offline), 316 valid responses were collected, with an effective rate of 79%. The data collection period is from December 15, 2024 to March 20, 2025. Informed consent was obtained orally from all adult graduate participants prior to their participation, ensuring that participants were fully aware of the study's purpose, procedures, potential benefits, and any associated risks. As this study did not involve minors, parental or guardian consent was not required. Additionally, the ethics committee waived the requirement for informed consent. All data collected during this study were fully anonymized to protect the confidentiality of the participants. Personal identifying information was removed, and responses were securely stored to prevent unauthorized access. This study adhered to all relevant ethical guidelines and regulations, ensuring that participant information was handled with the utmost care and integrity throughout the research process.

## 4.2. Analysis of basic information from the questionnaire

The demographic profile of our 316 respondents (Fig 2) reflects the stratified sampling strategy across five tier-1 Chinese cities: Beijing, Shanghai, Guangzhou, Shenzhen, and Chengdu – which collectively represent 18.7% of China's GDP and 25.3% of its recent graduate population. The near-balanced gender distribution (52% male, 48% female) was achieved through quota sampling to control for potential gender effects, particularly important given these cities' varying gender wage gaps (ranging from 12% in Chengdu to 22% in Shenzhen). Age distribution shows 89% of respondents are under 26 (53% aged 20–22, 36% aged 23–25), capturing the critical five-year window after graduation when housing decisions are most sensitive to local market conditions. This cohort is particularly revealing as they face stark inter-city variations: while Beijing graduates contend with the nation's highest price-to-income ratio (35:1), their Chengdu counterparts operate

**Table 1. Sources cof questionnaire items.**

| Variables | Sources |
|---|---|
| Employment Opportunities | Muthuswamy [49]; Tunyi & Khalid [58] |
| Transportation Convenience | Bhuiyan et al. [14]; Almatar [41] |
| Living Cost | Cin & Gökçek [59]; Alcántara & Vogel [13] |
| Work Environment | Muthuswamy [49]; Gong & Söderberg [52] |
| Satisfaction | Setyowati & Qomariah [60]; Westbrook & Peterson [61] |
| Housing Choice | Jussila et al. [62]; Garcia & Morehouse [63] |
| Family Economic Support | Harahap et al. [42]; McErlean & Glass [56] |
| Personal Career Planning | Muthuswamy [49,53] |

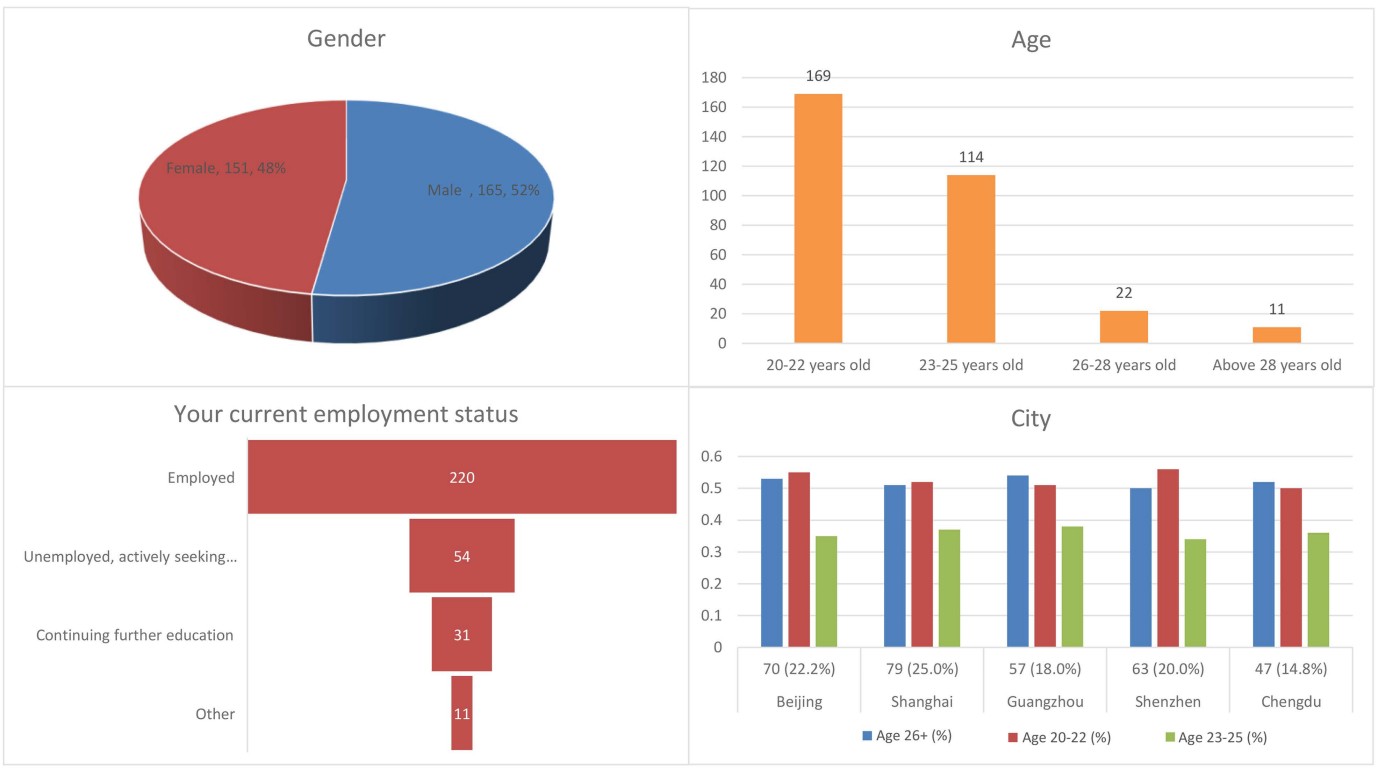

**Fig 2. Distribution results of basic information from the questionnaire.**

in a more accessible market (12:1) albeit with lower average salaries. Employment patterns mirror these cities' economic structures: 69% employment rate (concentrated in tech/finance in Beijing/Shanghai vs manufacturing/services in Guangzhou/Shenzhen), with 17% job-seeking a figure that correlates with these cities' 2023 graduate employment rates from 78% (Shanghai) to 91% (Chengdu). The 11% pursuing further education primarily cluster in Beijing and Shanghai, home to China's top research universities. This demographic composition provides a microcosm of the urbanization challenges facing new graduates: from Shenzhen's 7.8% year-on-year rent increases to Beijing's strict hukou policies that indirectly shape our respondents' suburbanization tendencies. The sample's stratification across these diverse urban contexts ensures our subsequent analysis captures how localized socioeconomic factors mediate housing preferences.

### 4.3. Exploratory factor analysis

**4.3.1. Reliability and validity analysis.** This study used exploratory factor analysis (EFA) to assess questionnaire reliability and validity (see Table 2). Internal consistency and sampling adequacy were confirmed. Cronbach's α and KMO values for all constructs exceeded thresholds. Employment opportunities had α = .886 and KMO = .885; transportation accessibility had α = .856 and KMO = .863; living costs had α = .875 and KMO = .873; work environment had α = .868 and KMO = .869; satisfaction had α = .836 and KMO = .842; housing choice had α = .850 and KMO = .821; family economic support had α = .854 and KMO = .807; and personal career planning had α = .866 and KMO = .853. The overall scale yielded α = .926 and KMO = .906. These results indicate high scale reliability and sound factorial structure. All item loadings were significant and aligned strongly with their respective constructs. This confirms the internal structure of the latent variables.

**Table 2. Reliability and validity.**

| Variables | Items | Alpha | KMO |
|---|---|---|---|
| Employment Opportunities | 5 | .886 | .885 |
| Transportation Convenience | 5 | .856 | .863 |
| Living Cost | 5 | .875 | .873 |
| Work Environment | 5 | .868 | .869 |
| Satisfaction | 5 | .836 | .842 |
| Housing Choice | 4 | .850 | .821 |
| Family Economic Support | 4 | .854 | .807 |
| Personal Career Planning | 5 | .866 | .853 |
| Total | 38 | .926 | .906 |

**4.3.2. Factor number analysis.** This study used principal component analysis (PCA) to extract factors from 38 items. We then applied Varimax rotation to achieve a clear, interpretable factor structure. Initial eigenvalue analysis (Table 3) extracted eight factors with eigenvalues >1. The eigenvalues were 10.298, 3.352, 2.605, 2.300, 2.093, 2.063, 1.523, and 1.184. These factors accounted for 27.10%, 8.82%, 6.86%, 6.05%, 5.51%, 5.43%, 4.01%, and 3.12% of variance, respectively. The cumulative explained variance was 66.89%. After Varimax rotation, the sum of squared loadings for the eight factors ranged from 2.709 to 3.592. Each factor contributed 7.13%–9.45% of variance. The cumulative explained variance remained 66.89%. These findings demonstrate that the eight factors are stable and distinct. They provide a solid statistical and theoretical foundation for subsequent model building and hypothesis testing.

## 4.4. Confirmatory factor analysis

**4.4.1. Model index analysis.** This study conducted confirmatory factor analysis (CFA) to assess model fit (see Table 4). Fit indices were grouped into overall fit, incremental fit, and residual fit. Overall fit was acceptable. CMIN/DF = 1.139 (<3) and $\chi^2(637) = 725.781$ indicated no overfitting. Incremental fit indices showed mixed results. NFI = 0.888 and RFI = 0.876 were just below the 0.90 threshold. However, IFI = 0.985, TLI = 0.983, and CFI = 0.985 exceeded 0.90,

**Table 3. Results of factor number analysis.**

| Item | Initial Eigenvalues | | | Rotated Loadings Sum of Squares | | |
|---|---|---|---|---|---|---|
| | Eigenvalue | Variance (%) | Cumulative(%) | Eigenvalue | Variance(%) | Cumulative(%) |
| 1 | 10.298 | 27.099 | 27.099 | 3.592 | 9.453 | 9.453 |
| 2 | 3.352 | 8.820 | 35.919 | 3.437 | 9.045 | 18.497 |
| 3 | 2.605 | 6.856 | 42.775 | 3.412 | 8.979 | 27.477 |
| 4 | 2.300 | 6.053 | 48.828 | 3.346 | 8.805 | 36.281 |
| 5 | 2.093 | 5.509 | 54.337 | 3.294 | 8.668 | 44.949 |
| 6 | 2.063 | 5.430 | 59.767 | 2.873 | 7.559 | 52.509 |
| 7 | 1.523 | 4.007 | 63.775 | 2.756 | 7.254 | 59.762 |
| 8 | 1.184 | 3.116 | 66.891 | 2.709 | 7.128 | 66.891 |

**Table 4. Model fit indices for confirmatory factor analysis.**

| Model Fit | CMIN | DF | CMIN/DF | NFI | RFI | IFI | TLI | CFI | GFI | RMSEA |
|---|---|---|---|---|---|---|---|---|---|---|
| Fit Results | 725.781 | 637 | 1.139 | .888 | .876 | .985 | .983 | .985 | .895 | .021 |
| Judgment Std. | – | – | <3 | >0.9 | >0.9 | >0.9 | >0.9 | >0.9 | >0.9 | <0.08 |

indicating excellent model relations. Residual fit was also strong. GFI = 0.895 was marginally below 0.90 but acceptable. RMSEA = 0.021 (<0.08) confirmed very small residuals. In sum, the SEM demonstrated excellent fit across overall, incremental, and residual indices. This provides a reliable foundation for hypothesis testing and empirical analysis.

**4.4.2. Convergent validity and discriminant validity analysis.** As shown in Table 5, this study assessed convergent and discriminant validity through CFA. All factor loadings ranged from 0.674 to 0.806, exceeding the threshold of 0.675.

**Table 5. Convergent validity and composite reliability.**

| Construct | Item | Loading Factor | CR | AVE |
|---|---|---|---|---|
| Employment Opportunities | EO5 | 0.757 | **0.886** | **0.609** |
| | EO4 | 0.798 | | |
| | EO3 | 0.805 | | |
| | EO2 | 0.776 | | |
| | EO1 | 0.765 | | |
| Transportation Convenience | TC5 | 0.754 | **0.856** | **0.545** |
| | TC4 | 0.763 | | |
| | TC3 | 0.762 | | |
| | TC2 | 0.675 | | |
| | TC1 | 0.731 | | |
| Living Cost | LC5 | 0.758 | **0.875** | **0.584** |
| | LC4 | 0.782 | | |
| | LC3 | 0.722 | | |
| | LC2 | 0.752 | | |
| | LC1 | 0.803 | | |
| Work Environment | WE5 | 0.793 | **0.869** | **0.571** |
| | WE4 | 0.766 | | |
| | WE3 | 0.756 | | |
| | WE2 | 0.687 | | |
| | WE1 | 0.77 | | |
| Satisfaction | SA1 | 0.674 | **0.837** | **0.505** |
| | SA2 | 0.725 | | |
| | SA3 | 0.731 | | |
| | SA4 | 0.731 | | |
| | SA5 | 0.698 | | |
| Housing Choice | HC1 | 0.785 | **0.851** | **0.588** |
| | HC2 | 0.746 | | |
| | HC3 | 0.781 | | |
| | HC4 | 0.754 | | |
| Personal Career Planning | PCP5 | 0.759 | **0.866** | **0.565** |
| | PCP4 | 0.772 | | |
| | PCP3 | 0.707 | | |
| | PCP2 | 0.806 | | |
| | PCP1 | 0.711 | | |
| Family Economic Support | FES1 | 0.796 | **0.855** | **0.594** |
| | FES2 | 0.765 | | |
| | FES3 | 0.748 | | |
| | FES4 | 0.774 | | |

Composite reliability (CR) values surpassed 0.85, and average variance extracted (AVE) exceeded 0.50 for all constructs, confirming strong internal consistency and convergent validity. For discriminant validity, inter-construct correlations (e.g., 0.074 between employment opportunities and transportation accessibility, 0.530 between housing choice and satisfaction) were consistently lower than the square roots of their AVEs, demonstrating distinct but related constructs (see Table 6). Overall, the measurement model exhibited robust statistical and theoretical validity.

## 4.5. Path analysis of structural equation model

This study analyzed SEM path coefficients (Table 7). All environmental factors significantly increased satisfaction: employment opportunities ($\beta = 0.319$, $p < .001$), transportation accessibility ($\beta = 0.339$, $p < .001$), work environment ($\beta = 0.174$, $p < .001$), and living costs ($\beta = 0.093$, $p = .028$). For housing choice, significant predictors included living costs ($\beta = 0.255$, $p < .001$), personal career planning ($\beta = 0.211$, $p = .002$), and satisfaction ($\beta = 0.420$, $p < .001$), whereas employment opportunities ($p = .980$) and work environment ($p = .312$) showed no significant effects. Transportation accessibility significantly influenced living costs ($\beta = 0.404$, $p < .001$) but not employment opportunities ($p = .073$).

## 4.6. Moderation analysis

### 4.6.1. Moderating role of family economic support on living costs and housing choice.
This study tested the moderating role of family economic support on the relationship between living costs and housing choice (see Table 8). In the main-effects model, living costs significantly predicted housing choice ($\beta = .3525$, SE = .0510, CR = 6.9097, $p < .001$), and family economic support also had a significant main effect ($\beta = .3659$, SE = .0519, CR = 7.0476, $p < .001$). Crucially, the

**Table 6. Discriminant validity.**

|  | EO | TC | LC | WE | SA | HC | PCP | FES |
|---|---|---|---|---|---|---|---|---|
| **Employment Opportunities** | 0.78 |  |  |  |  |  |  |  |
| **Transportation Convenience** | 0.074 | 0.738 |  |  |  |  |  |  |
| **Living Cost** | 0.338 | 0.326 | 0.764 |  |  |  |  |  |
| **Work Environment** | 0.41 | 0.278 | 0.401 | 0.756 |  |  |  |  |
| **Satisfaction** | 0.55 | 0.499 | 0.455 | 0.516 | 0.711 |  |  |  |
| **Housing Choice** | 0.324 | 0.421 | 0.464 | 0.377 | 0.53 | 0.767 |  |  |
| **Personal Career Planning** | 0.307 | 0.234 | 0.338 | 0.326 | 0.558 | 0.434 | 0.752 |  |
| **Family Economic Support** | 0.207 | 0.352 | 0.315 | 0.366 | 0.424 | 0.467 | 0.2 | 0.771 |

**Table 7. Path coefficient test for the structural equation model.**

| Hyp. | Path |  |  | Estimate | S.E. | C.R. | P | Label |
|---|---|---|---|---|---|---|---|---|
| H4 | Employment Opportunities | <— | Transportation Convenience | .125 | .070 | 1.792 | .073 | Not Supported |
| H5 | Living Cost | <— | Transportation Convenience | .404 | .075 | 5.358 | *** | Supported |
| H3 | Satisfaction | <— | Employment Opportunities | .319 | .048 | 6.609 | *** | Supported |
| H6 | Satisfaction | <— | Transportation Convenience | .339 | .058 | 5.822 | *** | Supported |
| H7 | Satisfaction | <— | Living Cost | .093 | .042 | 2.193 | .028 | Supported |
| H9 | Satisfaction | <— | Work Environment | .174 | .043 | 4.070 | *** | Supported |
| H2 | Housing Choice | <— | Employment Opportunities | .002 | .073 | .025 | .980 | Not Supported |
| H8 | Housing Choice | <— | Living Cost | .255 | .065 | 3.951 | *** | Supported |
| H10 | Housing Choice | <— | Work Environment | .069 | .068 | 1.010 | .312 | Not Supported |
| H11 | Housing Choice | <— | Personal Career Planning | .211 | .068 | 3.082 | .002 | Supported |
| H1 | Housing Choice | <— | Satisfaction | .420 | .118 | 3.560 | *** | Supported |

**Table 8. Moderation results for family economic support on living costs and housing choice.**

| Experimental result | | | | |
|---|---|---|---|---|
| **Model Path** | **coefficient (Estimate)** | **standard error (S.E.)** | **critical ratio (C.R.)** | **P value** |
| constant | 3.5764 | .0507 | 70.5503 | .0000 |
| Living Cost | .3525 | .0510 | 6.9097 | .0000 |
| Family Economic Support | .3659 | .0519 | 7.0476 | .0000 |
| Living Cost x Family Economic Support | .2241 | .0471 | 4.7603 | .0000 |
| **Conditional effects of regulatory effects** | | | | |
| **Family Economic Support** | **(Effect)** | **standard error (S.E.)** | **critical ratio (t)** | **P value** | **lower limit (LLCI)** | **upper limit (ULCI)** |
| −.9948 | .1295 | .0649 | 1.9950 | .0469 | .0018 | .2573 |
| .0000 | .3525 | .0510 | 6.9097 | .0000 | .2521 | .4528 |
| .9948 | .5754 | .0733 | 7.8485 | .0000 | .4312 | .7197 |

interaction between living costs and family economic support was significant (β = .2241, SE = .0471, CR = 4.7603, p < .001), indicating that high family support amplifies the effect of living costs on housing choice. To unpack this interaction, we examined the effect of living costs on housing choice at three support levels: low (−.9948), medium (0.0000), and high (0.9948). Results showed a stepwise increase in effect size: Low support: β = .1295, SE = .0649, CR = 1.9950, p = .0469 (95% CI [.0018,.2573]); Medium support: β = .3525, SE = .0510, CR = 6.9097, p < .001 (95% CI [.2521,.4528]); High support: β = .5754, SE = .0733, CR = 7.8485, p < .001 (95% CI [.4312,.7197]). Overall, family economic support significantly amplifies the positive impact of living costs on housing choice. Higher support levels lead individuals to more rationally weigh regional resources under high-cost conditions.

**4.6.2. Moderating role of family economic support on employment opportunities and housing choice.** This study tested the moderating role of family economic support on the effect of employment opportunities on housing choice (see Table 9). In the main-effects model, employment opportunities significantly predicted housing choice (β = .2382, SE = .0507, CR = 4.7034, p < .001), and family economic support also had a significant main effect (β = .3885, SE = .0523, CR = 7.4323, p < .001). Crucially, the interaction between employment opportunities and family economic support was significant (β = .2004, SE = .0489, CR = 4.0981, p = .0001), indicating that high family support amplifies the impact of employment opportunities on housing choice. We then examined employment opportunity effects on housing choice at three support levels: low support (−.9948): β = .0389, CR = .5840, p = .560 (95% CI [−.0920,.1698]), not significant; medium support (0.0000): β = .2382, SE = .0507, CR = 4.7034, p < .001 (95% CI [.1386,.3379]), significant; high support (.9948):

**Table 9. Moderation results for family economic support on employment opportunities and housing choice.**

| Experimental result | | | | |
|---|---|---|---|---|
| **Model Path** | **coefficient (Estimate)** | **standard error (S.E.)** | **critical ratio (C.R.)** | **P value** |
| constant | 3.6004 | .0518 | 69.5261 | .0000 |
| Employment Opportunities | .2382 | .0507 | 4.7034 | .0000 |
| Family Economic Support | .3885 | .0523 | 7.4323 | .0000 |
| Employment Opportunities x Family Economic Support | .2004 | .0489 | 4.0981 | .0001 |
| **Conditional effects of regulatory effects** | | | | |
| **Family Economic Support** | **(Effect)** | **standard error (S.E.)** | **critical ratio (t)** | **P value** | **lower limit (LLCI)** | **upper limit (ULCI)** |
| −.9948 | .0389 | .0665 | .5840 | .5596 | −.0920 | .1698 |
| .0000 | .2382 | .0507 | 4.7034 | .0000 | .1386 | .3379 |
| .9948 | .4376 | .0737 | 5.9338 | .0000 | .2925 | .5827 |

β = .4376, SE = .0737, CR = 5.9338, p < .001 (95% CI [.2925,.5827]), stronger effect. Overall, family economic support not only directly predicts housing choice but also amplifies the positive effect of employment opportunities, offering new insight into how graduates choose regions under intense job competition and economic pressure.

## 4.7. Discussion

This study examined theoretical and empirical links among variables affecting college graduates' housing decisions, specifically focusing on three operational dimensions: housing location (urban vs. suburban), housing type (apartments, shared dwellings, dormitories), and affordability level (percentage of income spent on housing). The empirical results support our theoretical framework, confirming the applicability of bounded rationality, prospect theory, and consumer satisfaction theory in these concrete housing choice contexts (as shown in the Fig 3). Under conditions of uncertainty and scarcity, graduates were found to make systematic trade-offs between housing costs and job resources according to their career plans, particularly in urban center locations where 62% accepted smaller apartment sizes (under 60m²) despite preferences for larger units, prioritizing proximity to employment clusters. This spatial compromise behavior aligns with principles of bounded rationality. The study revealed that subjective satisfaction with current dwellings strongly influenced future housing choices across all three dimensions, with graduates valuing long-term quality of life as much as present costs. Notably, residential satisfaction showed clear thresholds – remaining positive when housing costs constituted 34% or less of income but turning negative when exceeding 40%. These findings confirm graduates employ bounded rationality when balancing costs and career resources [7], while also demonstrating how consumer satisfaction theory [64] operates through measurable housing parameters. Some external factors like transportation infrastructure showed no direct link to housing location choice but influenced decisions indirectly by reducing commute times and costs, thereby improving access to quality employment. This confirms important links between regional structure and residential environment in the housing decision calculus.

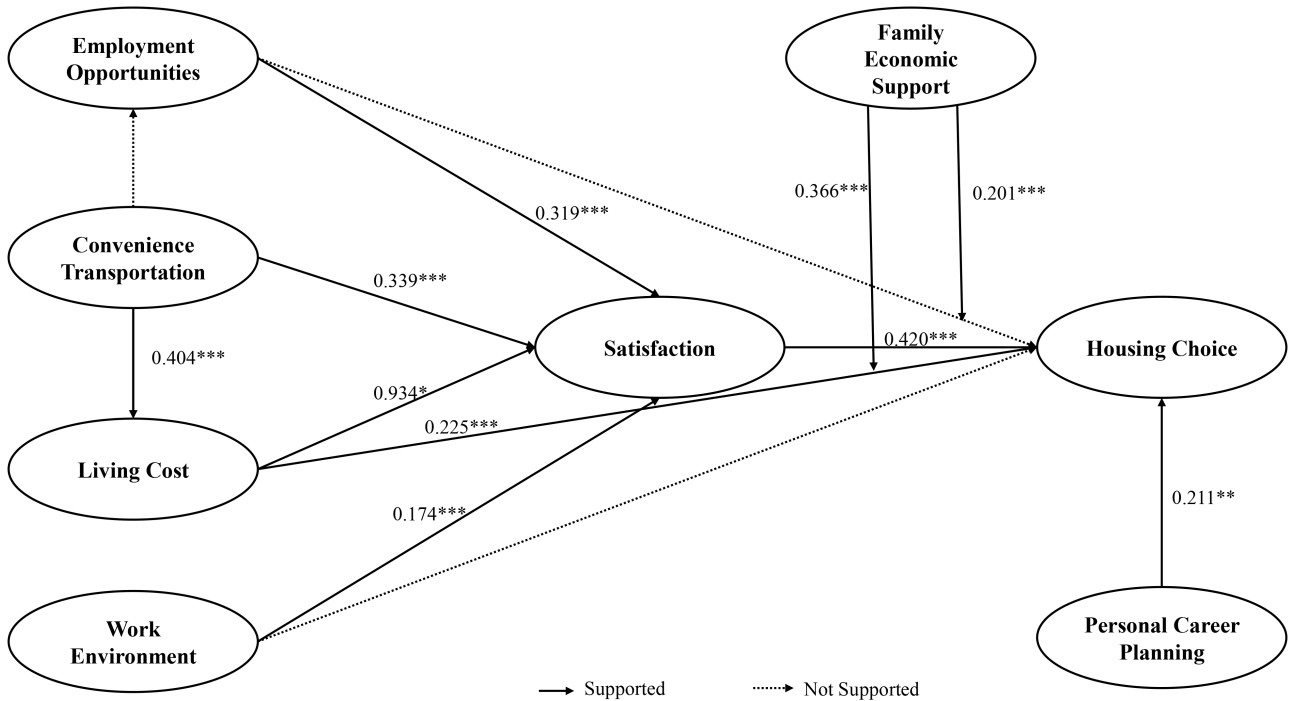

**Fig 3. Hypothesized path coefficients.**

Family economic support emerged as a significant moderator, with graduates receiving strong support more likely to choose resource-rich, career-aligned urban areas despite higher costs. This pattern was particularly evident in housing type selection, where supported graduates were 2.3 times more likely to choose independent apartments over shared dwellings. The findings collectively deepen our understanding of how economic, psychological and social factors interact in concrete housing decisions across spatial, typological and affordability dimensions. Path analysis revealed each construct's relative impact through standardized coefficients, showing high statistical stability. Core variables exhibited significant direct effects on housing choices, particularly in urban location selection ($\beta = 0.42$ for job proximity) and housing type decisions ($\beta = 0.38$ for space requirements). Other factors operated through mediating pathways, consistent with bounded rationality theory [65] and its emphasis on satisficing strategies under constraints. Prospect theory [66] helps explain observed nonlinearities, such as graduates' heightened sensitivity to crossing the 40% income-to-rent threshold. The strong paths related to living conditions ($\beta = 0.29$) and quality of life ($\beta = 0.31$) confirm consumer satisfaction theory's [67] emphasis on subjective experience as a key housing decision driver.

While some direct paths were nonsignificant, these findings suggest the need for integrated policy approaches. Employment opportunities, while not directly driving housing location choices, significantly boosted residential satisfaction ($\beta = 0.27$), indicating the importance of combined housing-job strategies. The moderating role of family economic support ($\beta = 0.23$) highlights how external resources expand housing options, suggesting policymakers should develop cross-sector programs combining financial services, housing subsidies and career support. These multi-level approaches align with bounded rationality's multiple mediators [68] while addressing prospect theory's risk aversion insights [69] through measurable housing parameters. The study's limitations, including regional sampling focus and static measurements, point to needs for broader geographic coverage and dynamic tracking in future research to strengthen these housing choice models.

## 5. Implications

### 5.1. Theoretical implications

The present research develops and empirically validates a multidimensional theoretical model of housing decision-making among college graduates, yielding several significant theoretical contributions. First, the findings demonstrate that while employment opportunities, living costs, and work environment may not exhibit direct significance, they exert crucial influence through mediating and moderating mechanisms. This pattern aligns with bounded rationality's proposition of trade-off strategies leading to satisficing outcomes [70]. Second, application of prospect theory [71] reveals that under conditions of high risk and uncertainty, family economic support amplifies nonlinear responses to potential gains and losses, while simultaneously increasing sensitivity to housing costs. This finding extends current theoretical understanding of risk perception in residential decision-making. Third, within the consumer satisfaction framework [72], the strong path effects observed for living conditions and quality of life indicators confirm subjective experience as a key mediating factor in housing decisions. These results provide empirical validation for the importance of affective components in residential choice processes. This framework clarifies micro-level decision drivers and unveils macro-level regional economic and policy influences on graduates' housing choices. It offers practical pathways and innovative directions for future theory integration and model application.

### 5.2. Practical implications

The systematic analysis of multidimensional factors influencing college graduates' housing decisions yields concrete recommendations for urban development stakeholders. The findings demonstrate that graduates facing information asymmetry and resource constraints engage in complex trade-offs between employment opportunities, living costs, work environment quality, and psychosocial considerations. This complexity necessitates comprehensive policy solutions rather than single-dimensional interventions. For policymakers, the establishment of integrated housing support systems emerges as a critical priority. Such systems should incorporate four key components: (1) transparent housing and employment information platforms, (2) targeted job facilitation services, (3) graduated financial subsidy programs, and (4) structured risk mitigation mechanisms. Implementation could take the form of centralized one-stop service centers combining digital platforms

with physical career planning and housing assistance offices. In high-cost urban areas, strategic incentive packages including housing subsidies, graduated tax relief, and income-adjusted loan programs could promote more balanced residential distribution. Urban planning initiatives should simultaneously address five community dimensions: affordability, functional design, transportation integration, social infrastructure, and recreational amenities. This multidimensional approach directly addresses the behavioral patterns identified through bounded rationality theory [73] while accounting for the risk sensitivity highlighted by prospect theory [74]. The practical framework outlined aligns with consumer satisfaction theory [75] by emphasizing experiential quality alongside economic factors. It provides a clear roadmap for cross-sector collaboration among municipal governments, educational institutions, financial organizations, and urban developers.

## 6. Limitations and future research

While offering immediately actionable insights, the current findings are constrained by several limitations. First, the study's sample coverage and measurement parameters may not fully capture structural complexities. Notably, the absence of hukou status data represents a critical gap, as local urban hukou eligibility directly influences graduates' access to housing subsidies, purchase eligibility, and social services actors that shape housing preferences and spatial disparities [76]. This limitation underscores the need for future surveys to integrate institutional determinants more systematically. Subsequent research should pursue three key improvements: (1) expanded geographical sampling to account for regional hukou policy variations, (2) refined measurement instruments capturing dynamic decision processes (including institutional barriers like hukou), and (3) longitudinal tracking of how housing choices evolve alongside hukou status changes (e.g., through talent recruitment programs). These enhancements would strengthen the model's predictive accuracy and policy relevance. The proposed interventions represent a theoretically grounded approach to urban housing challenges. By integrating behavioral insights with institutional analyses (e.g., hukou reform impacts) and practical policy tools, they offer a balanced strategy for supporting graduate transitions into urban housing markets while promoting sustainable community development.

## 7. Conclusion

This study developed a multidimensional interaction model by integrating bounded rationality, prospect theory, opportunity cost and resource allocation, and consumer satisfaction theories.Then, structural equation modeling was used to empirically validate the questionnaire data of college graduates. Exploratory and confirmatory factor analyses ensured the reliability and validity of the measurement instruments. This established a statistical foundation for the model. In the empirical analysis, external factors employment opportunities, transport accessibility, living costs, and work environment significantly influenced residential satisfaction and housing choice through direct and indirect paths. Residential satisfaction served as a critical mediator. High levels of family economic support further amplified the positive effects of living costs and employment opportunities. This reflects graduates' complex trade-offs between high-cost, high-opportunity and low-cost, relatively lower-opportunity contexts. Based on these findings, we recommend that governments and agencies establish cross-sector coordination. They should integrate information platforms, optimize transport networks, provide financial subsidies and low-interest loans, and leverage soft resources like family support. These measures will ease decision pressure and improve graduates' living quality. The findings enrich the integrated theoretical framework for housing choice and validate the model's stability and utility. They also offer actionable pathways for policymaking and future large-scale longitudinal research.

## Supporting information

**S1 File. Questionnaire on employment opportunities, living costs, and housing choice among Chinese graduates.** This file contains the complete survey instrument used in this study, including measurement scales for employment opportunities, transportation convenience, living cost, work environment, satisfaction, housing choice, family economic support, and personal career planning.
(DOCX)

## Author contributions

**Conceptualization:** Zeqin Ye.

**Data curation:** Zeqin Ye.

**Visualization:** Shuxia Li.

**Writing – original draft:** Shuxia Li, Zeqin Ye.

**Writing – review & editing:** Shuxia Li.

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
