## [Decision Letter · Decision Letter 0]

27 Jun 2025

PONE-D-25-22012Choosing Between City and Suburb: How Urbanization Shapes Graduates' Housing PreferencesPLOS ONE

Dear Dr. Li,

Thank you for submitting your manuscript to PLOS ONE. After careful consideration, we feel that it has merit but does not fully meet PLOS ONE’s publication criteria as it currently stands. Therefore, we invite you to submit a revised version of the manuscript that addresses the points raised during the review process.

We look forward to receiving your revised manuscript.

Kind regards,

Tianheng Shu, PhD

Academic Editor

PLOS ONE

**Journal Requirements:**

1. When submitting your revision, we need you to address these additional requirements. Please ensure that your manuscript meets PLOS ONE's style requirements, including those for file naming. The PLOS ONE style templates can be found at https://journals.plos.org/plosone/s/file?id=wjVg/PLOSOne_formatting_sample_main_body.pdf and https://journals.plos.org/plosone/s/file?id=ba62/PLOSOne_formatting_sample_title_authors_affiliations.pdf 2. Please update your submission to use the PLOS LaTeX template. The template and more information on our requirements for LaTeX submissions can be found at http://journals.plos.org/plosone/s/latex. 3. We note that your Data Availability Statement is currently as follows: All relevant data are within the manuscript and its Supporting Information files. Please confirm at this time whether or not your submission contains all raw data required to replicate the results of your study. Authors must share the “minimal data set” for their submission. PLOS defines the minimal data set to consist of the data required to replicate all study findings reported in the article, as well as related metadata and methods (https://journals.plos.org/plosone/s/data-availability#loc-minimal-data-set-definition). For example, authors should submit the following data: - The values behind the means, standard deviations and other measures reported;- The values used to build graphs;- The points extracted from images for analysis. Authors do not need to submit their entire data set if only a portion of the data was used in the reported study. If your submission does not contain these data, please either upload them as Supporting Information files or deposit them to a stable, public repository and provide us with the relevant URLs, DOIs, or accession numbers. For a list of recommended repositories, please see https://journals.plos.org/plosone/s/recommended-repositories. If there are ethical or legal restrictions on sharing a de-identified data set, please explain them in detail (e.g., data contain potentially sensitive information, data are owned by a third-party organization, etc.) and who has imposed them (e.g., an ethics committee). Please also provide contact information for a data access committee, ethics committee, or other institutional body to which data requests may be sent. If data are owned by a third party, please indicate how others may request data access. 4. One of the noted authors is a group or consortium. In addition to naming the author group, please list the individual authors and affiliations within this group in the acknowledgments section of your manuscript. Please also indicate clearly a lead author for this group along with a contact email address. 5. Please include your full ethics statement in the ‘Methods’ section of your manuscript file. In your statement, please include the full name of the IRB or ethics committee who approved or waived your study, as well as whether or not you obtained informed written or verbal consent. If consent was waived for your study, please include this information in your statement as well.

**Additional Editor Comments:**

The manuscript has been thoroughly reviewed by three reviewers, who have highlighted the need for specific clarifications to improve its quality. I encourage you to carefully review the detailed feedback provided in the review reports and address each comment comprehensively to enhance the overall quality of your manuscript.

Reviewers' comments:

Reviewer's Responses to Questions

**Comments to the Author**

1. Is the manuscript technically sound, and do the data support the conclusions?

Reviewer #1: Yes

Reviewer #2: Partly

Reviewer #3: Yes

2. Has the statistical analysis been performed appropriately and rigorously? 

Reviewer #1: Yes

Reviewer #2: Yes

Reviewer #3: Yes

3. Have the authors made all data underlying the findings in their manuscript fully available?

Reviewer #1: Yes

Reviewer #2: No

Reviewer #3: No

4. Is the manuscript presented in an intelligible fashion and written in standard English?

Reviewer #1: Yes

Reviewer #2: Yes

Reviewer #3: Yes

5. Review Comments to the Author

**Reviewer #1: **1. Some critical issues have been emphasized; however, some strong research gaps should be clarified

2. Statistics should be combined and made shorter

3. This study has reviewed several studies; however, some of the latest related studies should be added

**Reviewer #2:** This paper employs structural equation modeling to explore the factors influencing graduates' housing choices. While the introduction and literature review sections are relatively well-written, the data, methods, and findings sections require major revisions.

• [Survey Design Clarity] The paper lacks a clear explanation of the survey design. What is the target population? Are the participants college graduates from a single city, a province, or the entire country? If the target population comprises graduates from a single city, the authors should provide relevant socioeconomic background information about the city (e.g., population size, urbanization trends, socioeconomic indicators, major industries) to place the results in context more effectively.

• [Definition of College Graduates] How are "college graduates" defined in this study? Are they individuals who graduated within the past year? How many different colleges were the survey participants drawn from?

• [Inclusion of Hukou Status] The hukou status of college graduates is a crucial structural factor affecting their housing choices, but the paper does not address this. Depending on the study site, whether recent graduates have a local urban hukou can influence their access to social services, housing purchase options, and other life aspects that are directly or indirectly connected to housing choices and life satisfaction. If the authors do not have this piece of information, this should be mentioned as a limitation in the discussion section.

• [Questionnaire Items] It would be beneficial to include the sources of the questionnaire items and the exact wording of the survey questions (e.g., how is housing choice measured?). This information could be provided in the supplemental materials.

• [Result Interpretation] The results section needs more detailed and clear explanations. For instance, what do the authors mean by "housing choice"? Does it pertain to housing location (urban/suburban), housing type (apartments, townhouses, dormitories), or affordability level (percentage of income spent on housing)? Because the paper does not clearly operationalize concepts and explain the data collection process, the interpretation of results remains vague and lacks substantial meaning. Consequently, the findings are less useful for providing a deeper understanding of the topic or guiding policy decision-making.

**Reviewer #3: **1.Please reduce the number of keywords to 3-6 to align with journal guidelines and improve discoverability, retaining only those most central to your study.

2.While the manuscript presents an interesting approach to studying housing choice through satisfaction mediation, the theoretical justification for selecting satisfaction as the mediating variable requires stronger foundation. The authors should more thoroughly ground this choice in established theories like Expectation Confirmation Theory or Cognitive-Affective Theory, while also addressing why satisfaction is more appropriate than alternative psychological mechanisms. Regarding the independent variables, the current selection appears somewhat limited and would benefit from a clearer explanation of the inclusion criteria, particularly why other potentially relevant factors were excluded. The claim of constructing a multidimensional theoretical framework isn't fully substantiated in the current analysis - while statistical relationships are demonstrated, the integration of economic, social and psychological dimensions needs deeper theoretical articulation. The authors should either strengthen this theoretical integration by incorporating interdisciplinary perspectives or more explicitly acknowledge the current limitations of their framework.

3.The theoretical model appears to be merely a synthesis of existing literature without demonstrating original theoretical derivation or empirical validation. The authors must explicitly clarify the model's construction logic - whether derived deductively from established theories or inductively through empirical evidence - to establish its scholarly contribution.

6. PLOS authors have the option to publish the peer review history of their article (what does this mean?). If published, this will include your full peer review and any attached files.

Reviewer #1: No

Reviewer #2: No

Reviewer #3: No

---

## [Author Response · Author response to Decision Letter 1]

16 Aug 2025

Dear Editors and Reviewers,

We sincerely appreciate the opportunity to revise our manuscript and are grateful for the reviewers’ thoughtful comments. We have carefully addressed all concerns, and the revisions are highlighted in the resubmitted files.

We believe the manuscript has been significantly improved and hope it now meets PLOS ONE’s standards. Thank you for your time and consideration.

Sincerely,

Shuxia Li

Concord University College Fujian Normal University

---

## [Decision Letter · Decision Letter 1]

20 Oct 2025

Choosing Between City and Suburb: How Urbanization Shapes Graduates' Housing Preferences

PONE-D-25-22012R1

Dear Dr. Li,

We’re pleased to inform you that your manuscript has been judged scientifically suitable for publication and will be formally accepted for publication once it meets all outstanding technical requirements.

Kind regards,

Tianheng Shu, PhD

Academic Editor

PLOS ONE

Additional Editor Comments (optional):

Reviewers' comments:

Reviewer's Responses to Questions

**Comments to the Author**

1. If the authors have adequately addressed your comments raised in a previous round of review and you feel that this manuscript is now acceptable for publication, you may indicate that here to bypass the “Comments to the Author” section, enter your conflict of interest statement in the “Confidential to Editor” section, and submit your "Accept" recommendation.

Reviewer #1: All comments have been addressed

Reviewer #2: All comments have been addressed

2. Is the manuscript technically sound, and do the data support the conclusions?

Reviewer #1: Yes

Reviewer #2: (No Response)

3. Has the statistical analysis been performed appropriately and rigorously? 

Reviewer #1: Yes

Reviewer #2: (No Response)

4. Have the authors made all data underlying the findings in their manuscript fully available?

Reviewer #1: Yes

Reviewer #2: (No Response)

5. Is the manuscript presented in an intelligible fashion and written in standard English?

Reviewer #1: Yes

Reviewer #2: (No Response)

6. Review Comments to the Author

Reviewer #1: (No Response)

Reviewer #2: (No Response)

7. PLOS authors have the option to publish the peer review history of their article (what does this mean?). If published, this will include your full peer review and any attached files.

Reviewer #1: No

Reviewer #2: No

---

## [Editor Report · Acceptance letter]

PONE-D-25-22012R1

PLOS ONE

Dear Dr. Li,

I'm pleased to inform you that your manuscript has been deemed suitable for publication in PLOS ONE. Congratulations! Your manuscript is now being handed over to our production team.

Kind regards,

on behalf of

Dr. Tianheng Shu

Academic Editor

PLOS ONE